# Good Semi-supervised VAE Requires Tighter Evidence Lower Bound

## Abstract

Semi-supervised learning approaches based on generative models have now encountered 3 challenges: (1) The two-stage training strategy is not robust. (2) Good semi-supervised learning results and good generative performance can not be obtained at the same time. (3) Even at the expense of sacrificing generative performance, the semi-supervised classification results are still not satisfactory. To address these problems, we propose **O**ne-stage **S**emi-su**P**ervised **O**ptimal **T**ransport VAE (OSPOT-VAE), a one-stage deep generative model that theoretically unifies the generation and classification loss in one ELBO framework and achieves a tighter ELBO by applying the optimal transport scheme to the distribution of latent variables. We show that with tighter ELBO, our OSPOT-VAE surpasses the best semi-supervised generative models by a large margin across many benchmark datasets. For example, we reduce the error rate from 14.41% to 6.11% on Cifar-10 with 4k labels and achieve state-of-the-art performance with 25.30% on Cifar-100 with 10k labels. We also demonstrate that good generative models and semi-supervised results can be achieved simultaneously by OSPOT-VAE.

## 1 Introduction

The rise of deep neural networks has led to breakthroughs in computer vision, natural language processing, and many other domains. Most of these models are trained on large labeled datasets via supervised learning. However, in many scenarios, although it is easy to acquire a large amount of the original data, obtaining corresponding labels is often very costly or even infeasible. Semi-supervised learning (Thomas, 2009) is proposed to address this problem by training classifiers with sufficient unlabeled data and a small fraction of labeled data.

Recent works on semi-supervised learning can be grouped into three categories: (1) disagreement based learning via data perturbation (Miyato et al., 2019) and consistency enforcing (Verma et al., 2019), (2) metric learning (Wu et al., 2018), (3) generative approaches via generative adversarial network (GAN) (Springenberg, 2016) and variational autoencoder (VAE) (Kingma et al., 2014). Compared with the first two categories, generative approaches have great advantages in interpretability. Based on the latent variable assumption (Doersch, 2016), the generative model has an explicit variational inference form, so it can learn the marginal probability distribution of the raw data as well as the conditional distribution of the latent variables given the input data, which makes predictions more reasonable. Besides, generative approaches not only learn the required classification representations, but also capture the semantics-disentangled factors that generate the data, making it easier to generalize to different tasks (Narayanaswamy et al., 2017).

However, in practice, semi-supervised generative approaches often encounter three major challenges: (1) The two-stage training process is not robust. Semi-supervised VAE (Kingma et al., 2014) needs to be trained carefully with a two-stage hierarchical strategy, while the training process of GAN is a two-stage adversarial game (Chrysos et al., 2019). (2) Good semi-supervised learning results and good generative performance can not be obtained at the same time. In GAN, good semi-supervised learning performance will lead to a mismatch between the generated results and the real data distribution (Dai et al., 2017). While in VAE, the evidence lower bound (ELBO) objective is irrelevant to the classification loss, making it difficult to learn from the labels directly (Narayanaswamy et al., 2017). (3) Even at the expense of sacrificing generative performance, the semi-supervised classification results are still not satisfactory. In practice, disagreement-based meth-

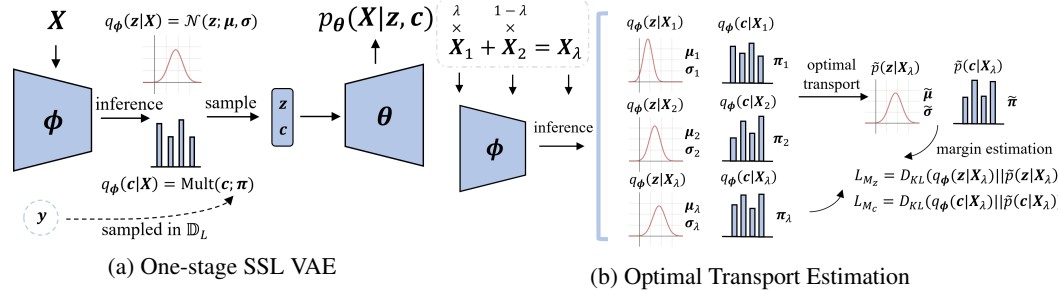

(a) One-stage SSL VAE

(b) Optimal Transport Estimation

Figure 1: The schematic of OSPOT-VAE

ods (Xie et al., 2019; Berthelot et al., 2019) have dramatically improved the state-of-the-art results on several standard datasets, surpassing generative approaches by a large margin. These challenges naturally raise a question: *What limits the performance of generative approaches in semi-supervised learning?*

In this work, we propose **O**ne-stage **S**emi-su**P**ervised **O**ptimal **T**ransport VAE (OSPOT-VAE) to address these challenges, which consists of two improvements: (1) a one-stage semi-supervised VAE model that unifies the generation and classification loss in one ELBO framework. (2) an estimation of the margin between true log-likelihood and the ELBO that exports a tighter evidence lower bound by applying optimal transport (Ambrosio & Gigli, 2013) scheme to the distribution of latent variables.

Our model has the following contributions:

- We show that OSPOT-VAE can be well trained with a direct one-stage strategy.
- We show that OSPOT-VAE can achieve both good generative performance and semi-supervised learning results simultaneously on a series of benchmark datasets.
- We point out that it is the large margin between the ELBO and the log-likelihood of the input data that limits the performance of semi-supervised VAE. Besides, we evaluate this assumption across many standard datasets and show that with the proposed tighter ELBO, OSPOT-VAE surpasses the best semi-supervised generative models by a large margin and achieves state-of-the-art performance on Cifar-100 with 10k labels.

## 2 SEMI-SUPERVISED LEARNING METHODS

In supervised learning (SL), we are facing with training data that appears as input-target pairs $(\mathbf{X}, \mathbf{y}) \in \mathbb{D}_L$ sampled from an unknown distribution $p(\mathbf{X}, \mathbf{y})$. Our goal is to learn a function $f(\mathbf{X}; \phi)$ parameterized by $\phi$ that makes the correct inference $\mathbf{y}$ for unseen samples from $p(\mathbf{X})$. While in semi-supervised learning (SSL), we can obtain an extra collection of unlabeled data $\mathbf{X} \in \mathbb{D}_U$ sampled from the same distribution $p(\mathbf{X})$. We hope to leverage the data from both $\mathbb{D}_L$ and $\mathbb{D}_U$ to achieve a more accurate model than what would have been obtained by only using $\mathbb{D}_L$.

In this section, we review some existing methods for SSL. We mainly focus on those who have reached state-of-the-art results, as well as generative approaches which are strongly connected with our model; the more comprehensive overview is beyond the scope of this paper, we refer readers to (Oliver et al., 2018).

### 2.1 DISAGREEMENT BASED LEARNING

Disagreement-based learning refers to the general approaches of imposing disagreement among multiple learners on the same task or multiple predictions from a single learner. By eliminating the disagreement, we can enforce the generalization of the model on unseen data. A common technique for creating disagreement is data augmentation, which applies transformations or perturbations on the input data and leaves class semantics unchanged. For $\mathbf{X} \in \mathbb{D}_U$, loss term can be derived as

$$\|f(\text{Augment}(\mathbf{X}); \phi) - f(\mathbf{X}; \phi)\|_2^2 \tag{1}$$

where the Augment($\mathbf{X}$) is a stochastic function which can be obtained by image transformation (Xie et al., 2019), virtual adversarial training (Miyato et al., 2019), or mixup method (Verma et al. 2018;

Berthelot et al. 2019). Another disagreement construction technique is to train multiple learners on the same dataset and utilize the loss

$$\|f(\mathbf{X}; \boldsymbol{\phi}_1) - f(\mathbf{X}; \boldsymbol{\phi}_2)\|_2^2 \tag{2}$$

to enforce the predictive consistency of different models, for example, "Mean Teacher" (Tarvainen & Valpola, 2017) and "Teacher Graph" (Luo et al., 2018). The generalization of the models gets enhanced.

## 2.2 GENERATIVE APPROACHES

In generative approaches, input $\mathbf{X}$ is supposed to have corresponding continuous and discrete latent variables, which we denote by $\mathbf{z}$ and $\mathbf{c}$ respectively.

**Feature matching (FM) GANs** (Salimans et al., 2016; Dai et al., 2017) apply GANs to semi-supervised learning on K-classification tasks by specifying a (K+1)-class objective for the discriminator. Instead of binary classification, true samples are classified into the first K classes respectively and fake samples are classified into the (K+1)-th class. This target function achieves strong empirical results by matching the generator distribution with true data distribution and improves semi-supervised classification performance.

**Semi-supervised VAEs** (Kingma et al. 2014; Narayanaswamy et al. 2017) construct a probabilistic model parameterized by $\boldsymbol{\theta}$ and $\boldsymbol{\phi}$ that respectively describe the generation and inference process between $\mathbf{X}$ and latent variables $\mathbf{z}$, $\mathbf{c}$. The generation process of $\mathbf{X}$ by $\mathbf{z}$ and $\mathbf{c}$ is :

$$p(\mathbf{z}) = \mathcal{N}(\boldsymbol{z}; \mathbf{0}, \boldsymbol{I}); \qquad p(\mathbf{c}) = \text{Mult}(\mathbf{c}; K, \boldsymbol{\pi}); \qquad p_{\boldsymbol{\theta}}(\mathbf{X}|\mathbf{z}, \mathbf{c}) = f(\mathbf{X}; \mathbf{z}, \mathbf{c}, \boldsymbol{\theta}) \tag{3}$$

where $\text{Mult}(K, \boldsymbol{\pi})$ is the multinomial distribution with class $K$ and parameter $\boldsymbol{\pi}$. $f(\mathbf{X}; \mathbf{z}, \mathbf{c}, \boldsymbol{\theta})$ is a suitable likelihood function, e.g. a Bernoulli or Gaussian distribution, parameterized by a non-linear transformation of the latent variables $\mathbf{z}$ and $\mathbf{c}$. The class label $\mathbf{y}$ is treated as $\mathbf{c}$ if given. For the inference process, with the following hypothesis

$$q_{\boldsymbol{\phi}}(\mathbf{z}, \mathbf{c}|\mathbf{X}) = q_{\boldsymbol{\phi}}(\mathbf{z}|\mathbf{X})q_{\boldsymbol{\phi}}(\mathbf{c}|\mathbf{X}); \qquad p(\mathbf{z}, \mathbf{c}|\mathbf{X}) = p(\mathbf{z}|\mathbf{X})p(\mathbf{c}|\mathbf{X}); \qquad p(\mathbf{z}, \mathbf{c}) = p(\mathbf{z})p(\mathbf{c}) \tag{4}$$

evidence lower bound (ELBO) is used as objective to predict the posterior distribution of latent variables as follows (see Appendix A.1 for proof):

$$\log p(\mathbf{X}) \geq \mathbb{E}_{q_{\boldsymbol{\phi}}(\mathbf{z}, \mathbf{c}|\mathbf{X})}[\log p_{\boldsymbol{\theta}}(\mathbf{X}|\mathbf{z}, \mathbf{c})] - D_{\text{KL}}(q_{\boldsymbol{\phi}}(\mathbf{z}|\mathbf{X})\|p(\mathbf{z})) - D_{\text{KL}}(q_{\boldsymbol{\phi}}(\mathbf{c}|\mathbf{X})\|p(\mathbf{c})) = \text{ELBO} \tag{5}$$

For the likelihood $\log p(\mathbf{X})$ is infeasible, VAE maximizes its evidence lower bound instead, which derives the negative ELBO loss function $\mathcal{L}(\mathbf{X}; \boldsymbol{\phi}, \boldsymbol{\theta}) = -\text{ELBO}$. The predictions for the classification label $\mathbf{y}$ can be obtained from the inferred posterior distribution $q_{\boldsymbol{\phi}}(\mathbf{c}|\mathbf{X})$. When class label $\mathbf{y}$ is not given, missing label sampling technique is used to sample from $q_{\boldsymbol{\phi}}(\mathbf{c}|\mathbf{X})$ as

$$\mathbb{E}_{q_{\boldsymbol{\phi}}(\mathbf{c}|\mathbf{X})}f(\mathbf{X}; \mathbf{c}, \boldsymbol{\theta}) = \sum_{k=1}^{K} q_{\boldsymbol{\phi}}(\mathbf{y}_k|\mathbf{X})f(\mathbf{X}; \mathbf{y}_k, \boldsymbol{\theta}) \tag{6}$$

Here $\mathbf{y}_k$ represents a one-hot vector with 1 in k-th dimension. Utilizing this sampling method, the algorithmic complexity of VAE is proportional to K so it is computationally inefficient. Note that in objective (4), the label predictive distribution $q_{\boldsymbol{\phi}}(\mathbf{c}|\mathbf{X})$ only contributes to the generative performance. To remedy this, existing models simply add a cross-entropy loss to the negative ELBO loss such that the distribution $q_{\boldsymbol{\phi}}(\mathbf{c}|\mathbf{X})$ can also learn classification rules from the labeled data. The extended objective loss is

$$\min_{\boldsymbol{\phi}, \boldsymbol{\theta}} \mathbb{E}_{\mathbf{X} \sim \mathbb{D}_U} \mathcal{L}(\mathbf{X}; \boldsymbol{\phi}, \boldsymbol{\theta}) + \mathbb{E}_{(\mathbf{X}, \mathbf{y}) \sim \mathbb{D}_L}[\mathcal{L}(\mathbf{X}, \mathbf{c} = \mathbf{y}; \boldsymbol{\phi}, \boldsymbol{\theta}) - \log q_{\boldsymbol{\phi}}(\mathbf{y}|\mathbf{X})] \tag{7}$$

**Two-stage Training Strategy**: In practice, (Kingma et al., 2014) finds that directly training the one-stage objective (7) will lead to a bad semi-supervised learning result, so a two-stage training strategy is proposed to improve the model. The two-stage training strategy consists of two parts, M1 and M2. M1 means to learn a new continuous latent representation $\mathbf{z}_1$ first, and M2 means to train a semi-supervised model (7) with the embedding $\mathbf{z}_1$ from M1 instead of the raw data $\mathbf{X}$. This M1+M2 strategy builds a deep VAE with two layers of random variables: $p_{\boldsymbol{\theta}}(\mathbf{X}, \mathbf{z}_1, \mathbf{z}_2, \mathbf{c}) = p_{\boldsymbol{\theta}}(\mathbf{X}|\mathbf{z}_1)p_{\boldsymbol{\theta}}(\mathbf{z}_1|\mathbf{z}_2, \mathbf{c})p(\mathbf{z}_2)p(\mathbf{c})$, which can dramatically improve the performance of the inference $q_{\boldsymbol{\phi}}(\mathbf{c}|\mathbf{X})$ but is not robust in training. Moreover, GAN's training process can also be considered as two-stage with generator and discriminator competing with each other, and the two-stage adversarial game adds the difficulty in training.

# 3 ONE-STAGE SEMI-SUPERVISED OPTIMAL TRANSPORT VAE

In this section, we introduce our semi-supervised VAE framework, OSPOT-VAE. Firstly, we derive a one-stage loss function that unifies the generation and classification loss under one ELBO without introducing any additional auxiliary loss items like (7). Then, we analyze a phenomenon that good ELBO values do not guarantee good semi-supervised performance and propose the optimal transport estimation to deal with it. At last, combining the two parts, we give the detailed algorithm of OSPOT-VAE and discuss some problems in model optimization.

## 3.1 ONE-STAGE SEMI-SUPERVISED VAE

Following the notations and assumptions $(3, 4)$ in Section 2.2, we derive our one-stage semi-supervised VAE. With the empirical distribution $p_{emp}(\mathbf{X}; \mathbb{D}) = \frac{1}{|\mathbb{D}|} \sum_{\mathbf{X}' \in \mathbb{D}} \mathbf{1}_{\mathbf{X}=\mathbf{X}'}$, we utilize the decomposition in (Zhao et al., 2017) and rewrite the second part of (5) into (proof in Appendix A.2)

$$\mathbb{E}_{p_{emp}(\mathbf{X})} D_{\mathrm{KL}}(q_\phi(\mathbf{z}|\mathbf{X})\|p(\mathbf{z})) = \mathbf{I}_{q_\phi}(\mathbf{X}; \mathbf{z}) + D_{\mathrm{KL}}(q_\phi(\mathbf{z})\|p(\mathbf{z})) \geq \mathbf{I}_{q_\phi}(\mathbf{X}; \mathbf{z}) \qquad (8)$$

where $q_\phi(\mathbf{z}) = \frac{1}{|\mathbb{D}|} \sum_{\mathbf{X} \in \mathbb{D}} q_\phi(\mathbf{z}|x)$ and $\mathbf{I}_{q_\phi}(\mathbf{X}; \mathbf{z})$ is the mutual information between $\mathbf{X}$ and $\mathbf{z}$. The left part of (8) equals to 0 when $\mathbf{X}$ and $\mathbf{z}$ are independent. This is undesirable, so $\mathbf{I}_{q_\phi}(\mathbf{X}; \mathbf{z})$ can be regarded as the lower bound of controlled mutual information. The continuous variables in (8) can be easily extend to discrete variables $\mathbf{c}$. We can use $\mathbf{I}_{\mathbf{z}}$ and $\mathbf{I}_{\mathbf{c}}$ to denote the controlled information capacity and derive the objective for the unlabeled dataset $\mathbb{D}_U$

$$\mathcal{L}_{\mathbb{D}_U}(\mathbf{X}; \boldsymbol{\theta}, \boldsymbol{\phi}) = \mathbb{E}_{q_\phi(\mathbf{z},\mathbf{c}|\mathbf{X})}[-\log p_{\boldsymbol{\theta}}(\mathbf{X}|\mathbf{z}, \mathbf{c})] + \beta_{\mathbf{z}}|D_{\mathrm{KL}}(q_\phi(\mathbf{z}|\mathbf{X})\|p(\mathbf{z}) - \mathbf{I}_{\mathbf{z}}|$$
$$+ \beta_{\mathbf{c}}|D_{\mathrm{KL}}(q_\phi(\mathbf{c}|\mathbf{X})\|p(\mathbf{c})) - \mathbf{I}_{\mathbf{c}}| \qquad (9)$$

where $\beta, \mathbf{I}_{\mathbf{z}}, \mathbf{I}_{\mathbf{c}}$ are all hyper-parameters forcing the KL divergence term to match the mutual information capacities of $\mathbf{z}$ and $\mathbf{c}$.

For the labeled subset $\mathbb{D}_L$, instead of directly employing class label $\mathbf{y}$ as sampled $\mathbf{c}$, we view it as the parameter of the true posterior distribution, i.e. $p(\mathbf{c}|\mathbf{X}) = \mathrm{Mult}(\mathbf{c}; K, \mathbf{y})$ and derive the following one-stage ELBO form:

$$\log p(\mathbf{X}) = \log \mathbb{E}_{q_\phi(\mathbf{z}|\mathbf{X}),p(\mathbf{c}|\mathbf{X})} \frac{p(\mathbf{X}, \mathbf{z}, \mathbf{c})}{q_\phi(\mathbf{z}|\mathbf{X})p(\mathbf{c}|\mathbf{X})} \geq \mathbb{E}_{q_\phi(\mathbf{z}|\mathbf{X}),p(\mathbf{c}|\mathbf{X})} \log \frac{p(\mathbf{X}, \mathbf{z}, \mathbf{c})}{q_\phi(\mathbf{z}|\mathbf{X})p(\mathbf{c}|\mathbf{X})}$$

$$= \mathbb{E}_{q_\phi(\mathbf{z}|\mathbf{X}),p(\mathbf{c}|\mathbf{X})}[\log p(\mathbf{X}|\mathbf{z}, \mathbf{c}) + \log \frac{p(\mathbf{z})p(\mathbf{c})}{q_\phi(\mathbf{z}|\mathbf{X})p(\mathbf{c}|\mathbf{X})}] = \mathbb{E}_{q_\phi(\mathbf{z}|\mathbf{X}),p(\mathbf{c}|\mathbf{X})} \log p(\mathbf{X}|\mathbf{z}, \mathbf{c}) \quad (10)$$

$$- D_{\mathrm{KL}}(q_\phi(\mathbf{z}|\mathbf{X})\|p(\mathbf{z})) - D_{\mathrm{KL}}(p(\mathbf{c}|\mathbf{X})\|q_\phi(\mathbf{c}|\mathbf{X})) + \mathbb{E}_{p(\mathbf{c}|\mathbf{X})} \log \frac{p(\mathbf{c})}{q_\phi(\mathbf{c}|\mathbf{X})}$$

Notice that $D_{\mathrm{KL}}(p(\mathbf{c}|\mathbf{X})\|q_\phi(\mathbf{c}|\mathbf{X}))$ is equal to the common cross-entropy loss for y is a one-hot vector. In this respect, the margin between $p(\mathbf{c}|\mathbf{X})$ and $q_\phi(\mathbf{c}|\mathbf{X})$ can be significantly small when the suitable optimization method is chosen. This allows us to utilize the approximation $p(\mathbf{c}|\mathbf{X}) \approx q_\phi(\mathbf{c}|\mathbf{X})$ to modify $\mathbb{E}_{p(\mathbf{c}|\mathbf{X})} \log \frac{p(\mathbf{c})}{q_\phi(\mathbf{c}|\mathbf{X})}$ in (10), resulting in a consist ELBO with $\mathcal{L}_{\mathbb{D}_U}(\mathbf{X}; \boldsymbol{\theta}, \boldsymbol{\phi})$:

$$\mathbb{E}_{p(\mathbf{c}|\mathbf{X})} \log \frac{p(\mathbf{c})}{q_\phi(\mathbf{c}|\mathbf{X})} \approx^{(\text{when } p(\mathbf{c}|\mathbf{X}) \approx q_\phi(\mathbf{c}|\mathbf{X}))} \mathbb{E}_{q_\phi(\mathbf{c}|\mathbf{X})} \log \frac{p(\mathbf{c})}{q_\phi(\mathbf{c}|\mathbf{X})} = D_{\mathrm{KL}}(q_\phi(\mathbf{c}|\mathbf{X})\|p(\mathbf{c})) \quad (11)$$

Combining the ELBO form (10) of $\mathbb{D}_L$ with the mutual information decomposition (8) and the approximation (11), the new objective for semi-supervised VAE is:

$$\mathcal{L}_{\mathbb{D}_L}(\mathbf{X}, \mathbf{y}; \boldsymbol{\theta}, \boldsymbol{\phi}) = \mathbb{E}_{q_\phi(\mathbf{z}|\mathbf{X}),p(\mathbf{c}|\mathbf{X})}[-\log p_{\boldsymbol{\theta}}(\mathbf{X}|\mathbf{z}, \mathbf{c})] + \beta_{\mathbf{z}}|D_{\mathrm{KL}}(q_\phi(\mathbf{z}|\mathbf{X})\|p(\mathbf{z})) - \mathbf{I}_{\mathbf{z}}|$$
$$+ \beta_{\mathbf{c}}|D_{\mathrm{KL}}(q_\phi(\mathbf{c}|\mathbf{X})\|p(\mathbf{c})) - \mathbf{I}_{\mathbf{c}}| + D_{\mathrm{KL}}(p(\mathbf{c}|\mathbf{X})\|q_\phi(\mathbf{c}|\mathbf{X})) \qquad (12)$$

With (9) and (12), the objective for the entire dataset is now

$$\min_{\boldsymbol{\phi},\boldsymbol{\theta}} \mathbb{E}_{\mathbf{X} \sim p_{emp}(\mathbf{X};\mathbb{D}_U)} \mathcal{L}_{\mathbb{D}_U}(\mathbf{X}; \boldsymbol{\theta}, \boldsymbol{\phi}) + \mathbb{E}_{(\mathbf{X},\mathbf{y}) \sim p_{emp}((\mathbf{X},\mathbf{y});\mathbb{D}_L)} \mathcal{L}_{\mathbb{D}_L}(\mathbf{X}, \mathbf{y}; \boldsymbol{\theta}, \boldsymbol{\phi}) \qquad (13)$$

This one-stage objective with a simple approximate transformation (9) unifies the generation loss as well as the target of SSL and results in improved performance of semi-supervised learning, which we demonstrate in Section 4.1.

---

**Algorithm 1** Optimal transport estimation ingests a batch of observation $\mathbf{X}$ as well as the representation $q_\phi(\mathbf{z}|\mathbf{X}), q_\phi(\mathbf{c}|\mathbf{X})$ inferred from the original VAE and returns the estimation of the margin $D_{\mathrm{KL}}(q_\phi(\mathbf{z}|\mathbf{X})\|p(\mathbf{z}|\mathbf{X}))$ and $D_{\mathrm{KL}}(q_\phi(\mathbf{c}|\mathbf{X})\|p(\mathbf{c}|\mathbf{X}))$.

**Input:**
  Batch of observation $\mathbf{X}$ sampled from $p_{emp}(\mathbf{X})$;
  Inferred parameter $(\boldsymbol{\mu}, \mathrm{diag}(\boldsymbol{\sigma}^2))$ of $q_\phi(\mathbf{z}|\mathbf{X}) = \mathcal{N}(\mathbf{z}; \boldsymbol{\mu}, \mathrm{diag}(\boldsymbol{\sigma}^2))$;
  Inferred parameter $\boldsymbol{\pi}$ of $q_\phi(\mathbf{c}|\mathbf{X}) = \mathrm{Mult}(\mathbf{c}; K, \boldsymbol{\pi})$;
  Hyperparameter $\alpha$ for mixup vicinal distribution $p_{mixup}(\mathbf{X})$
**Output:**
  $\tilde{\mathbf{X}}$ sampled from $p_{mixup}(\mathbf{X})$;
  Estimation $L_{M_\mathbf{z}}$ of the margin $D_{\mathrm{KL}}(q_\phi(\mathbf{z}|\tilde{\mathbf{X}})\|p(\mathbf{z}|\tilde{\mathbf{X}}))$;
  Estimation $L_{M_\mathbf{c}}$ of the margin $D_{\mathrm{KL}}(q_\phi(\mathbf{c}|\tilde{\mathbf{X}})\|p(\mathbf{c}|\tilde{\mathbf{X}}))$
  1: $\mathbf{X}', \boldsymbol{\pi}', \boldsymbol{\mu}', \boldsymbol{\sigma}'^2 = \mathrm{RandomPermutation}(\mathbf{X}, \boldsymbol{\pi}, \boldsymbol{\mu}, \boldsymbol{\sigma}^2)$
  2: $\tilde{\mathbf{X}} = \lambda * \mathbf{X} + (1 - \lambda) * \mathbf{X}', \lambda \in \beta(\alpha, \alpha)$
  3: $\tilde{\boldsymbol{\pi}} = \mathrm{OptimalTransportC}(\boldsymbol{\pi}, \boldsymbol{\pi}', \lambda)$
  4: $(\tilde{\boldsymbol{\mu}}, \tilde{\boldsymbol{\sigma}}^2) = \mathrm{OptimalTransportZ}((\boldsymbol{\mu}, \boldsymbol{\sigma}^2), (\boldsymbol{\mu}', \boldsymbol{\sigma}'^2), \lambda)$
  5: $q_\phi(\mathbf{z}|\tilde{\mathbf{X}}), q_\phi(\mathbf{c}|\tilde{\mathbf{X}}) = \mathrm{VAE}(\tilde{\mathbf{X}})$
  6: $\tilde{p}(\mathbf{z}|\tilde{\mathbf{X}}) = \mathcal{N}(\mathbf{z}; \tilde{\boldsymbol{\mu}}, \mathrm{diag}(\tilde{\boldsymbol{\sigma}}^2))$
  7: $\tilde{p}(\mathbf{c}|\tilde{\mathbf{X}}) = \mathrm{Mult}(\mathbf{c}; K, \tilde{\boldsymbol{\pi}})$
  8: $L_{M_\mathbf{z}} = D_{\mathrm{KL}}(q_\phi(\mathbf{z}|\tilde{\mathbf{X}})\|\tilde{p}(\mathbf{z}|\tilde{\mathbf{X}}))$
  9: $L_{M_\mathbf{c}} = D_{\mathrm{KL}}(q_\phi(\mathbf{c}|\tilde{\mathbf{X}})\|\tilde{p}(\mathbf{c}|\tilde{\mathbf{X}}))$
 10: **return** $\tilde{\mathbf{X}}, L_{M_\mathbf{z}}, L_{M_\mathbf{c}}$

---

### 3.2 OPTIMAL TRANSPORT ESTIMATION

To summarize the above, VAE aims to learn the useful representation $q_\phi(\mathbf{z}|\mathbf{X})$ and $q_\phi(\mathbf{c}|\mathbf{X})$ by reducing the KL divergence between the empirical distribution $p_{emp}(\mathbf{X})$ and the model marginal $p(\mathbf{X}) = \int_\mathbf{z} \int_\mathbf{c} p(\mathbf{X})p(\mathbf{z}|\mathbf{X})p(\mathbf{c}|\mathbf{X})d\mathbf{z}d\mathbf{c}$. Instead of minimizing $D_{\mathrm{KL}}(p_{emp}(\mathbf{X})\|p(\mathbf{X}))$ directly, VAE models use the expected ELBO mentioned in (5) as target via the following inequality

$$D_{\mathrm{KL}}(p_{emp}(\mathbf{X})\|p(\mathbf{X})) \leq H(p_{emp}(\mathbf{X})) - \mathbb{E}_{p_{emp}(\mathbf{X})}\mathrm{ELBO} \tag{14}$$

However, one phenomenon is that good ELBO values do not imply accurate inference. A typical example has been discussed in (Zhao et al., 2017). Here we mainly focus on the cause of this phenomenon and propose optimal transport estimation to alleviate this problem in semi-supervised learning. Following the work in (Rezende et al., 2014), we write down the closed form of the expected margin between true log-likelihood and ELBO as (proof in Appendix A.3):

$$\mathbb{E}_{p_{emp}(\mathbf{X})}[\log p(\mathbf{X}) - \mathrm{ELBO}] = \mathbb{E}_{p_{emp}(\mathbf{X})}[D_{\mathrm{KL}}(q_\phi(\mathbf{z}|\mathbf{X})\|p(\mathbf{z}|\mathbf{X})) + D_{\mathrm{KL}}(q_\phi(\mathbf{c}|\mathbf{X})\|p(\mathbf{c}|\mathbf{X}))] \tag{15}$$

Combined with the decomposition (8), training the expected ELBO target can only reduce the difference between marginal distributions $q_\phi(\mathbf{c}), q_\phi(\mathbf{z})$ and $p(\mathbf{c}), p(\mathbf{z})$. It means that even with a good ELBO, the margin $\mathbb{E}_{p_{emp}(\mathbf{X})}D_{\mathrm{KL}}(q_\phi(\mathbf{z}|\mathbf{X})\|p(\mathbf{z}|\mathbf{X}))$ and $\mathbb{E}_{p_{emp}(\mathbf{X})}D_{\mathrm{KL}}(q_\phi(\mathbf{c}|\mathbf{X})\|p(\mathbf{c}|\mathbf{X}))$ in (15) can still be large. In this scenario, the consistent optimization of ELBO will contribute no more to the semi-supervised classification performance. However, optimizing the margin in (15) directly is impossible, for $p(\mathbf{c}|\mathbf{X})$ and $p(\mathbf{z}|\mathbf{X})$ are unknown. To remedy this, we extend the empirically effective approximation in (Zhang et al., 2018) to our VAE framework with the form

$$\mathbb{E}_{p_{mixup}(\mathbf{X})}D_{\mathrm{KL}}(q_\phi(\mathbf{z}|\mathbf{X})\|p(\mathbf{z}|\mathbf{X})) \approx \mathbb{E}_{p_{emp}(\mathbf{X})}D_{\mathrm{KL}}(q_\phi(\mathbf{z}|\mathbf{X})\|p(\mathbf{z}|\mathbf{X}))(\alpha \to 0)$$
$$\mathbb{E}_{p_{mixup}(\mathbf{X})}D_{\mathrm{KL}}(q_\phi(\mathbf{c}|\mathbf{X})\|p(\mathbf{c}|\mathbf{X})) \approx \mathbb{E}_{p_{emp}(\mathbf{X})}D_{\mathrm{KL}}(q_\phi(\mathbf{c}|\mathbf{X})\|p(\mathbf{c}|\mathbf{X}))(\alpha \to 0) \tag{16}$$

where $p_{mixup}(\mathbf{X})$ is the mixup vicinal distribution (Zhang et al., 2018) and $\alpha$ is the related parameter. Then we propose optimal transport estimation to construct the estimations of $\mathbb{E}_{p_{mixup}(\mathbf{X})}D_{\mathrm{KL}}(q_\phi(\mathbf{z}|\mathbf{X})\|p(\mathbf{z}|\mathbf{X}))$ as well as $\mathbb{E}_{p_{mixup}(\mathbf{X})}D_{\mathrm{KL}}(q_\phi(\mathbf{z}|\mathbf{X})\|p(\mathbf{z}|\mathbf{X}))$ by applying optimal transport scheme to latent variables $\mathbf{z}$ and $\mathbf{c}$. The computation steps of optimal transport estimation are provided in Algorithm 1, and we present the details of the optimal transport scheme in the rest of this section.

---

**Algorithm 2** OSPOT-VAE training process with epoch $t$.

---

**Input:**

Batch of labeled pairs $(\mathbf{X}_L, \mathbf{y}_L) \in \mathbb{D}_L$, Batch of unlabeled examples $\mathbf{X}_U \in \mathbb{D}_U$;

ELBO hyperparameters: $\beta_{\mathbf{z}}, \beta_{\mathbf{c}}, \mathbf{I}_{\mathbf{z}}, \mathbf{I}_{\mathbf{c}} = \text{ELBOScheduler}(t)$;

Optimal transport estimation weights: $w_{M_{\mathbf{z}}}, w_{M_{\mathbf{c}}} = \text{WeightScheduler}(t)$;

Model parameters: $\boldsymbol{\theta}^{(t-1)}, \boldsymbol{\phi}^{(t-1)}$;

Model optimizer: SGD

**Output:**

Updated parameters: $\boldsymbol{\theta}^{(t)}, \boldsymbol{\phi}^{(t)}$

1: $L_L = \mathcal{L}_{\mathbb{D}_L}(\mathbf{X}_L, \mathbf{y}_L; \boldsymbol{\theta}^{(t-1)}, \boldsymbol{\phi}^{(t-1)}; \beta_{\mathbf{z}}, \beta_{\mathbf{c}}, \mathbf{I}_{\mathbf{z}}, \mathbf{I}_{\mathbf{c}})$

2: $L_U = \mathcal{L}_{\mathbb{D}_U}(\mathbf{X}_U; \boldsymbol{\theta}^{(t-1)}, \boldsymbol{\phi}^{(t-1)}; \beta_{\mathbf{z}}, \beta_{\mathbf{c}}, \mathbf{I}_{\mathbf{z}}, \mathbf{I}_{\mathbf{c}})$

3: $L_{M_{\mathbf{z}}}, L_{M_{\mathbf{c}}} = \text{OptimalTransportEstimation}(\mathbf{X}_U, q_{\boldsymbol{\phi}}(\mathbf{z}|\mathbf{X}_U), q_{\boldsymbol{\phi}}(\mathbf{c}|\mathbf{X}_U))$

4: $L = L_L + L_U + w_{M_{\mathbf{z}}} L_{M_{\mathbf{z}}} + w_{M_{\mathbf{c}}} L_{M_{\mathbf{c}}}$

5: $\boldsymbol{\theta}^{(t)}, \boldsymbol{\phi}^{(t)} = \text{SGD}(\boldsymbol{\theta}^{(t-1)}, \boldsymbol{\phi}^{(t-1)}, \frac{\partial L}{\partial \boldsymbol{\theta}}, \frac{\partial L}{\partial \boldsymbol{\phi}})$

6: **return** $\boldsymbol{\theta}^{(t)}, \boldsymbol{\phi}^{(t)}$

---

**Optimal Transport Scheme**: The mixup vicinal distribution can be understood as applying linear transport between the points $\mathbf{X}, \mathbf{X}' \in \mathbb{D}$, extending the original dataset with new points falling on one straight line $\tilde{\mathbf{X}} = \lambda * \mathbf{X} + (1 - \lambda) * \mathbf{X}', \lambda \in [0, 1]$. For $\tilde{\mathbf{X}}$, it is a natural thought that this linear transformation could associate with the shortest-path transport in the latent space. Based on this, we calculate the distributions $\tilde{p}(\mathbf{z}|\tilde{\mathbf{X}}), \tilde{p}(\mathbf{c}|\tilde{\mathbf{X}})$ of $\mathbf{z}, \mathbf{c}$ and consider them as the estimation of the true posterior distributions. Following the work of (Ambrosio & Gigli, 2013), the norm-2 based optimal transport scheme $\gamma(\mathbf{x}, \mathbf{y})$ between two distributions $p(\mathbf{x})$ and $p(\mathbf{y})$ satisfy:

$$\inf_{\gamma(\mathbf{x}, \mathbf{y})} \int_{\mathbf{x}} \int_{\mathbf{y}} \|\mathbf{x} - \mathbf{y}\|_2^2 \gamma(\mathbf{x}, \mathbf{y}) d\mathbf{x} d\mathbf{y}$$

$$\text{s.t.} \int_{\mathbf{y}} \gamma(\mathbf{x}, \mathbf{y}) d\mathbf{y} = p(\mathbf{x}); \int_{\mathbf{x}} \gamma(\mathbf{x}, \mathbf{y}) d\mathbf{x} = p(\mathbf{y}) \tag{17}$$

For the continuous variable $\mathbf{z} \sim \mathcal{N}(\boldsymbol{\mu}, \text{diag}(\boldsymbol{\sigma}^2))$ and discrete variable $\mathbf{c} \sim \text{Mult}(K, \boldsymbol{\pi})$, the following 2 propositions are proposed to calculate the shortest-path based on optimal transport scheme (see Appendix A.4 for proof).

**Proposition 3.1.** *The shortest-path derived from optimal transport scheme* (17) *between* $\mathbf{z}_1 \sim \mathcal{N}(\boldsymbol{\mu}_1, diag(\boldsymbol{\sigma}_1^2))$ *and* $\mathbf{z}_2 \sim \mathcal{N}(\boldsymbol{\mu}_2, diag(\boldsymbol{\sigma}_2^2))$ *with* $\lambda \in [0, 1]$ *is*

$$\tilde{\boldsymbol{\mu}} = \lambda \boldsymbol{\mu}_1 + (1 - \lambda) \boldsymbol{\mu}_2$$
$$\tilde{\boldsymbol{\sigma}} = \lambda \boldsymbol{\sigma}_1 + (1 - \lambda) \boldsymbol{\sigma}_2 \tag{18}$$

**Proposition 3.2.** *The shortest-path derived from KL divergence based optimal transport scheme between* $\mathbf{c}_1 \sim Mult(K, \boldsymbol{\pi}_1)$ *and* $\mathbf{c}_2 \sim Mult(K, \boldsymbol{\pi}_2)$ *with* $\lambda \in [0, 1]$ *is*

$$\tilde{\boldsymbol{\pi}} = \lambda \boldsymbol{\pi}_1 + (1 - \lambda) \boldsymbol{\pi}_2 \tag{19}$$

Algorithm 1 yields the optimal transport estimation of the margin in (15), which leads to a tighter ELBO. In Section 4.2, we demonstrate that with this tighter ELBO, the inference performance of semi-supervised VAE is significantly improved on many benchmark datasets.

## 3.3 OPTIMIZATION OF OSPOT-VAE

Combining one-stage semi-supervised VAE and optimal transport estimation, we can get the complete OSPOT-VAE model. The full OSPOT-VAE algorithm is provided in Algorithm 2, and a schematic is shown in Figure 1. Note that the conditions for the approximations used in Algorithm 1,2 satisfy (1) $q_{\boldsymbol{\phi}}(\mathbf{c}|\mathbf{X}) \approx p(\mathbf{c}|\mathbf{X})$ and (2) the VAE model has already achieved a good ELBO. Therefore, the warm-up schedule (Higgins et al., 2017) is used to set parameters $\mathbf{I}_z, \mathbf{I}_c, \beta_{\mathbf{z}}, \beta_{\mathbf{c}}$ and $w_{M_{\mathbf{z}}}, w_{M_{\mathbf{c}}}$. We list the details of "ELBOScheduler$(t)$" and "WeightScheduler$(t)$" in Appendix A.5.

In Algorithm 2, we apply stochastic gradient descent (SGD) as optimizer, which needs to calculate the gradient $\nabla_{\boldsymbol{\theta}, \boldsymbol{\phi}} L$. The target loss $L$ consists of KL divergence and the expected log-likelihood . The derivation of KL divergence part has a closed form, while calculating the gradient

| BackBone | Method | MNIST(100 labels) | SVHN(1k labels) |
|---|---|---|---|
| Same with M1+M2 | Disentangled VAE (Narayanaswamy et al., 2017) | 9.71($\pm$0.91) | 38.91($\pm$1.06) |
| | M1(Kingma et al., 2014) | 11.97($\pm$1.71) | 54.33($\pm$0.11) |
| | M1+M2(Kingma et al., 2014) | 3.33($\pm$0.14) | 36.02($\pm$0.10) |
| | **One-stage VAE** | **3.14**($\pm$0.19) | **27.38**($\pm$0.78) |

Table 1: One-stage VAE error rate in MNIST and SVHN.

| BackBone | Model category | Model | Cifar10(4k labels) |
|---|---|---|---|
| WRN-28-2 | Disagreement | Temporal Ensembling(TE) (Laine & Aila, 2017) | 16.37 |
| | | Mean Teacher(Tarvainen & Valpola, 2017) | 15.87 |
| | | VAT+EntMin(Miyato et al., 2019) | 13.13 |
| | | MixMatch(Berthelot et al., 2019) | **6.37** |
| | Generative | GS-BadGAN[†][*](Li et al., 2019) | 17.11 |
| | | **OSPOT-VAE** | **8.51**($\pm$0.32) |
| WRN-28-10 | Disagreement | AutoAugment(Cubuk et al., 2019) | 14.1 |
| | | Temporal Ensembing(Laine & Aila, 2017) | 12.16 |
| | | MixMatch[*](Berthelot et al., 2019) | **4.95** |
| | Generative | GS-BadGAN[†][*](Li et al., 2019) | 14.41 |
| | | GAN combine TE[‡][*](Wei et al., 2018) | 9.98 |
| | | **OSPOT-VAE** | **6.11**($\pm$0.34) |

Table 2: Error rate in Cifar10. † denotes the best semi-supervised generative approach result. ‡ denotes the model ensemble two categories. ∗ denotes the corresponding backbone is not exactly WideResNet (Zagoruyko & Komodakis, 2016), but belongs to one kind of its variations with a comparable amount of parameters.

of $\mathbb{E}_{q_\phi(\mathbf{z}|\mathbf{X}), q_\phi(\mathbf{c}|\mathbf{X})} \log p_\theta(\mathbf{X}|\mathbf{z}, \mathbf{c})$ is difficult. To this end, we follow the work of (Rezende et al., 2014) and (Jang et al., 2017), using the reparameterization trick as

$$\nabla_{\theta,\phi}\mathbb{E}_{q_\phi(\mathbf{z};\mathbf{X})} \log p_\theta(\mathbf{X}|\mathbf{z}) = \mathbb{E}_{\mathcal{N}(\epsilon;\mathbf{0},\mathbf{I})}\nabla_{\theta,\phi} \log p_\theta(\mathbf{X}|\boldsymbol{\mu} + \boldsymbol{\sigma} \cdot \boldsymbol{\epsilon})$$

$$\mathbb{E}_{\mathbf{Gumbel}(\epsilon;\mathbf{0},\mathbf{1})}\nabla_{\theta,\phi} \log p_\theta(\mathbf{X}|\text{Softmax}(\frac{\log \boldsymbol{\pi} + \boldsymbol{\epsilon}}{\tau})) \rightarrow \nabla_{\theta,\phi}\mathbb{E}_{q_\phi(\mathbf{c}|\mathbf{X})} \log p_\theta(\mathbf{X}|\mathbf{c})(\tau \rightarrow 0) \quad (20)$$

Note that with (20), the algorithmic complexity of one-stage semi-supervised VAE is independent with the class number $K$, making it easier to extend to large-scale classification tasks.

## 4 EXPERIMENTS

In this section, we demonstrate the 3 contributions of our OSPOT-VAE model with sufficient experiments on 4 standard SSL benchmark datasets, that is, MNIST, SVHN, Cifar10, and Cifar100. In Section 4.1, we show the validity of our one-stage semi-supervised VAE objective (13) by comparing with other one-stage and two-stage VAE models. Then, we evaluate the performance of OSPOT-VAE under "WideResNet"(Zagoruyko & Komodakis, 2016) backbone and compare with other state-of-the-art SSL models mentioned in Section 2. Besides, We provide an ablation study to verify the contribution of the optimal transport estimation. As an additional application, we show that good generative models and semi-supervised results can be obtained at the same time by OSPOT-VAE (Section 4.3). The source code is available at https://github.com/PaperCodeSubmission/OSPOT-VAE; more details are available in Appendix A.6.

### 4.1 ONE-STAGE SEMI-SUPERVISED VAE

We evaluate the effectiveness of the one-stage semi-supervised VAE objective on 2 standard benchmarks, MNIST and SVHN. As for baseline models, we consider two VAE-based SSL models, which are one-stage disentangled VAE (Narayanaswamy et al., 2017) and two-stage VAE(M1+M2) (Kingma et al., 2014). For fairness, except the target loss functions, all models use the same structure as is used in M1+M2 (Kingma et al., 2014). The results are presented in Table 1, and our model achieves the best performance.

Table 3: Error in Cifar100. † and ∗ have the same meaning as described in Table 2.

| BackBone | Model | Cifar100(4k labels) | Cifar100(10k labels) |
|---|---|---|---|
| WRN-28-2 | $\Pi$ − Model(Laine & Aila, 2017) | \ | 39.19 |
| | GS-BadGAN[†][*](Li et al., 2019) | 45.11 | 37.16 |
| | LP[*](Iscen et al., 2019) | 43.73 | 35.92 |
| | **OSPOT-VAE** | **40.58**($\pm$0.48) | **31.41**($\pm$0.21) |
| WRN-28-10 | MixMatch[*] (Berthelot et al., 2019) | \ | 25.88 |
| | **OSPOT-VAE** | **33.76**($\pm$0.53) | **25.30**($\pm$0.31) |

Table 4: Ablation study with SVHN, Cifar10, and Cifar100.

| Methods | SVHN(1k labels) | Cifar10(4k labels) | Cifar100(10k labels) |
|---|---|---|---|
| One-stage VAE | 10.53($\pm$0.17) | 18.26($\pm$0.51) | 38.62($\pm$0.67) |
| Optimal transport estimation (with encoder only) | 6.54($\pm$0.62) | 10.71($\pm$0.44) | 36.21($\pm$0.29) |
| OSPOT-VAE | **5.79**($\pm$0.15) | **8.51**($\pm$0.32) | **31.41**($\pm$0.21) |

## 4.2 OSPOT-VAE

We compare the results of OSPOT-VAE with two categories of state-of-the-art models mentioned in Section 2. In all experiments, we use the "WideResNet-28" model or other deep models with a comparable amount of parameters as the backbone. The results in Table 2,3 demonstrate that our model outperforms most of the existing methods and surpasses state-of-the-art semi-supervised generative models (Dai et al., 2017) by a large margin. Notice that recently, data-augmentation based method, MixMatch, (Berthelot et al., 2019) achieves the absolute state-of-the-art results in all benchmarks. It uses pre-designed sophisticated data augmentation strategies for different datasets and outperforms our model. We list its results fairly as a comparison, while OSPOT-VAE surpasses it in Cifar100 dataset.

**Ablation Study**: The OSPOT-VAE model consists of two parts: (1) a one-stage VAE objective and (2) an optimal transport estimation. In ablation study, we analyze the effect of each component in our model with the backbone "WideResNet-28-2". To study the independent effects of transport estimation, we combine it with the encoder part of OSPOT-VAE to build a classifier with loss function $L_{M_{\mathbf{c}}}$. The improved classification error rates in Table 4 show that, with optimal transport estimation, the posterior inference $q_{\phi}(\mathbf{c}|\mathbf{X})$ gets closer to the true distribution $p(\mathbf{c}|\mathbf{X})$. It indicates that our optimal transport estimation does reduce the gap between ELBO and the log-likelihood of the input data and yield a tighter ELBO, which leads to a better semi-supervised performance.

Table 5: Generative performance measured by ELBO with ELBO $\leq \log p(\mathbf{X})$

| Model | Cifar10 | Cifar100 |
|---|---|---|
| Pure VAE | $-226.25(\pm14.25)$ | $-1292.91(\pm1.10)$ |
| OSPOT-VAE | $-237.62(\pm6.27)$ | $-1271.82(\pm24.15)$ |

## 4.3 GENERATIVE PERFORMANCE

$\mathbb{E}_{p_{emp}(\mathbf{X})}$ELBO measures the margin between the true data distribution and the distribution learned by generation models (Doersch, 2016). By comparing the $\mathbb{E}_{p_{emp}(\mathbf{X})}$ value of pure unsupervised VAE and our semi-supervised VAE model under the same "WideResNet-28-2" backbone, we demonstrate that good generative models and semi-supervised results can be obtained at the same time in OSPOT-VAE. The results in Table 5 show that the data generative distribution learned by our OSPOT-VAE model is as good as the pure VAE model. Further generated results are available in Appendix A.7.

## 5 CONCLUSION

In this work, we pointed out that it was the large margin between ELBO and the true log-likelihood of the raw data that limits the performance of semi-supervised VAE. To this end, we introduced OSPOT-VAE, a one-stage generative model that unified the classification and generation objective and achieved a tighter ELBO by optimal transport estimation. We demonstrated our assertion through extensive experiments, and our semi-supervised results significantly outperform former state-of-the-art generative SSL methods by a large margin on Cifar10 and Cifar100.

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

## A APPENDIX

### A.1 BASIC INEQUALITY OF ELBO

**Proposition A.1.** *The Basic inequality of ELBO is*
$$\log p(\mathbf{X}) \geq \mathbb{E}_{q_\phi(\mathbf{z},\mathbf{c}|\mathbf{X})}[\log p_\theta(\mathbf{X}|\mathbf{z},\mathbf{c})] - D_{\mathrm{KL}}(q_\phi(\mathbf{z}|\mathbf{X})\|p(\mathbf{z})) - D_{\mathrm{KL}}(q_\phi(\mathbf{c}|\mathbf{X})\|p(\mathbf{c}))$$

**proof**

$$\log p(\mathbf{X}) = \log \int_{\mathbf{z},\mathbf{c}} q_\phi(\mathbf{z},\mathbf{c}|\mathbf{X}) \frac{p(\mathbf{X},\mathbf{z},\mathbf{c})}{q_\phi(\mathbf{z},\mathbf{c}|\mathbf{X})} = \log \mathbb{E}_{q_\phi(\mathbf{z},\mathbf{c}|\mathbf{X})} \frac{p(\mathbf{X},\mathbf{z},\mathbf{c})}{q_\phi(\mathbf{z},\mathbf{c}|\mathbf{X})}$$

$$\geq \mathbb{E}_{q_\phi(\mathbf{z},\mathbf{c}|\mathbf{X})} \log \frac{p(\mathbf{X},\mathbf{z},\mathbf{c})}{q_\phi(\mathbf{z},\mathbf{c}|\mathbf{X})}$$

$$= \int_{\mathbf{z},\mathbf{c}} q_\phi(\mathbf{z},\mathbf{c}|\mathbf{X}) \log \frac{p(\mathbf{z},\mathbf{c})}{q_\phi(\mathbf{z},\mathbf{c}|\mathbf{X})} + \int_{\mathbf{z},\mathbf{c}} q_\phi(\mathbf{z},\mathbf{c}|\mathbf{X}) \log p_\theta(\mathbf{X}|\mathbf{z},\mathbf{c})$$

$$= -D_{\mathrm{KL}}(q_\phi(\mathbf{z},\mathbf{c}|\mathbf{X})\|p(\mathbf{z},\mathbf{c})) + \mathbb{E}_{q_\phi(\mathbf{z},\mathbf{c}|\mathbf{X})}[\log p_\theta(\mathbf{X}|\mathbf{z},\mathbf{c})]$$

$$=^{\text{with assumption (3,4)}} \mathbb{E}_{q_\phi(\mathbf{z},\mathbf{c}|\mathbf{X})}[\log p_\theta(\mathbf{X}|\mathbf{z},\mathbf{c})] - D_{\mathrm{KL}}(q_\phi(\mathbf{z}|\mathbf{X})\|p(\mathbf{z})) - D_{\mathrm{KL}}(q_\phi(\mathbf{c}|\mathbf{X})\|p(\mathbf{c}))$$

$\square$

### A.2 DECOMPOSITION OF ELBO IN INFO-VAE

**Proposition A.2.** *The expected ELBO with empirical distribution satisfies*
$$\mathbb{E}_{p_{emp}(\mathbf{X})} D_{\mathrm{KL}}(q_\phi(\mathbf{z}|\mathbf{X})\|p(\mathbf{z})) = \mathbf{I}_{q_\phi}(\mathbf{X};\mathbf{z}) + D_{\mathrm{KL}}(q_\phi(\mathbf{z})\|p(\mathbf{z}))$$

**proof**

$$\mathbb{E}_{p_{emp}(\mathbf{X})} D_{\mathrm{KL}}(q_\phi(\mathbf{z}|\mathbf{X})\|p(\mathbf{z})) = \int_{\mathbf{X}} p_{emp}(\mathbf{X}) \int_{\mathbf{z}} q_\phi(\mathbf{z}|\mathbf{X}) \frac{q_\phi(\mathbf{z}|\mathbf{X})}{p(\mathbf{z})} d\mathbf{z}d\mathbf{X}$$

$$= \int_{\mathbf{X}} p_{emp}(\mathbf{X}) \int_{\mathbf{z}} q_\phi(\mathbf{z}|\mathbf{X}) \frac{q_\phi(\mathbf{z}|\mathbf{X})}{q_\phi(\mathbf{z})} d\mathbf{z}d\mathbf{X} + \int_{\mathbf{X}} p_{emp}(\mathbf{X}) \int_{\mathbf{z}} q_\phi(\mathbf{z}|\mathbf{X}) \frac{q_\phi(\mathbf{z})}{p(\mathbf{z})} d\mathbf{z}d\mathbf{X}$$

$$= \int_{\mathbf{X}} \int_{\mathbf{z}} q_\phi(\mathbf{z},\mathbf{X}) \log \frac{q_\phi(\mathbf{z},\mathbf{X})}{q_\phi(\mathbf{z})p_{emp}(\mathbf{X})} d\mathbf{z}d\mathbf{X} + D_{\mathrm{KL}}(q_\phi(\mathbf{z})\|p(\mathbf{z}))$$

$$= \mathbf{I}_{q_\phi}(\mathbf{X};\mathbf{z}) + D_{\mathrm{KL}}(q_\phi(\mathbf{z})\|p(\mathbf{z}))$$

$\square$

### A.3 THE EQUATION FORM OF ELBO

**Proposition A.3.** *The expected margin between the true log-likelihood* $\mathbb{E}_{p_{emp}(\mathbf{X})} \log p(\mathbf{X})$ *and* $\mathbb{E}_{p_{emp}(\mathbf{X})} ELBO$ *is*

$$\mathbb{E}_{p_{emp}(\mathbf{X})}[\log p(\mathbf{X}) - ELBO] = \mathbb{E}_{p_{emp}(\mathbf{X})}[D_{\mathrm{KL}}(q_\phi(\mathbf{z}|\mathbf{X})\|p(\mathbf{z}|\mathbf{X})) + D_{\mathrm{KL}}(q_\phi(\mathbf{c}|\mathbf{X})\|p(\mathbf{c}|\mathbf{X}))]$$

**proof**

We just need to prove the following equation

$$\log p(\mathbf{X}) - \mathrm{ELBO} = D_{\mathrm{KL}}(q_\phi(\mathbf{z}|\mathbf{X})\|p(\mathbf{z}|\mathbf{X})) + D_{\mathrm{KL}}(q_\phi(\mathbf{c}|\mathbf{X})\|p(\mathbf{c}|\mathbf{X}))$$

and the proof under assumption $(3, 4)$ is

$$\log p(\mathbf{X}) = \int_{\mathbf{z},\mathbf{c}} q_\phi(\mathbf{z}|\mathbf{X}) q_\phi(\mathbf{c}|\mathbf{X}) \log p(\mathbf{X}) d\mathbf{z} d\mathbf{c} = \int_{\mathbf{z},\mathbf{c}} q_\phi(\mathbf{z}|\mathbf{X}) q_\phi(\mathbf{c}|\mathbf{X}) \log \frac{p(\mathbf{X},\mathbf{z},\mathbf{c})}{p(\mathbf{z}|\mathbf{X})p(\mathbf{c}|\mathbf{X})} d\mathbf{z} d\mathbf{c}$$

$$= \mathbb{E}_{q_\phi(\mathbf{z}|\mathbf{X})q_\phi(\mathbf{c}|\mathbf{X})} \log \frac{p(\mathbf{X},\mathbf{z},\mathbf{c})}{q_\phi(\mathbf{z}|\mathbf{X})q_\phi(\mathbf{c}|\mathbf{X})} + \int_{\mathbf{z},\mathbf{c}} q_\phi(\mathbf{z}|\mathbf{X}) q_\phi(\mathbf{c}|\mathbf{X}) \log \frac{q_\phi(\mathbf{z}|\mathbf{X})q_\phi(\mathbf{c}|\mathbf{X})}{p(\mathbf{z}|\mathbf{X})p(\mathbf{c}|\mathbf{X})} \qquad \Box$$

$$= \mathrm{ELBO} + D_{\mathrm{KL}}(q_\phi(\mathbf{z}|\mathbf{X})\|p(\mathbf{z}|\mathbf{X})) + D_{\mathrm{KL}}(q_\phi(\mathbf{c}|\mathbf{X})\|p(\mathbf{c}|\mathbf{X}))$$

### A.4 OPTIMAL TRANSPORT SCHEME

**Proposition A.4.**

1. The shortest-path derived from optimal transport scheme (17) between $\mathbf{z}_1 \sim \mathcal{N}(\boldsymbol{\mu}_1, \mathrm{diag}(\boldsymbol{\sigma}_1^2))$ and $\mathbf{z}_2 \sim \mathcal{N}(\boldsymbol{\mu}_2, \mathrm{diag}(\boldsymbol{\sigma}_2^2))$ with $\lambda \in [0,1]$ is

$$\begin{aligned} \tilde{\boldsymbol{\mu}} &= \lambda\boldsymbol{\mu}_1 + (1-\lambda)\boldsymbol{\mu}_2 \\ \tilde{\boldsymbol{\sigma}} &= \lambda\boldsymbol{\sigma}_1 + (1-\lambda)\boldsymbol{\sigma}_2 \end{aligned} \tag{A.1}$$

**proof**

Utilizing the conclusions in Kuang & Tabak (2017), the closed form of optimal transport from one multi-normal distribution $\mathcal{N}(\mathbf{z}_1; \boldsymbol{\mu}_1, \boldsymbol{\Sigma}_1)$ to another normal distribution $\mathcal{N}(\mathbf{z}_2; \boldsymbol{\mu}_2, \boldsymbol{\Sigma}_2)$ is

$$\mathbf{z} \to \mathcal{T}(\mathbf{z}) = \boldsymbol{\mu}_2 + \mathbf{T}(\mathbf{z} - \boldsymbol{\mu}_1); \qquad \mathbf{T} = \boldsymbol{\Sigma}_1^{-\frac{1}{2}}(\boldsymbol{\Sigma}_1^{\frac{1}{2}}\boldsymbol{\Sigma}_2\boldsymbol{\Sigma}_1^{\frac{1}{2}})^{\frac{1}{2}}\boldsymbol{\Sigma}_1^{-\frac{1}{2}}$$

Utilize the diag matrix assumption, the optimal transport scheme with $\lambda$ is

$$\begin{aligned} \mathbf{z}_\lambda &= (1-\lambda)\mathbf{z}_1 + \lambda\mathcal{T}(\mathbf{z}_1) \\ \mathbf{T} &= diag(\boldsymbol{\sigma}_1/\boldsymbol{\sigma}_2) \end{aligned} \tag{A.2}$$

Utilize (A.2), we can get (A.1) as

$$\mathbf{z}_\lambda \sim \mathcal{N}(\tilde{\boldsymbol{\mu}}, diag(\tilde{\boldsymbol{\sigma}}^2)); \qquad \tilde{\boldsymbol{\mu}} = \lambda\boldsymbol{\mu}_1 + (1-\lambda)\boldsymbol{\mu}_2; \qquad \tilde{\boldsymbol{\sigma}} = \lambda\boldsymbol{\sigma}_1 + (1-\lambda)\boldsymbol{\sigma}_2 \quad \Box$$

2. The shortest-path derived from KL divergence based optimal transport scheme between $\mathbf{c}_1 \sim \mathrm{Mult}(K, \boldsymbol{\pi}_1)$ and $\mathbf{c}_2 \sim \mathrm{Mult}(K, \boldsymbol{\pi}_2)$ with $\lambda \in [0,1]$ is

$$\tilde{\boldsymbol{\pi}} = \lambda\boldsymbol{\pi}_1 + (1-\lambda)\boldsymbol{\pi}_2 \tag{A.3}$$

**proof**

As the definition (17) has no closed-form solution for multinomial distribution, we use KL divergence instead. The KL divergence based optimal transport target $\mathbf{c}_\lambda \sim \mathrm{Mult}(K, \tilde{\boldsymbol{\pi}})$ between two multinomial distribution $\mathbf{c}_1 \sim \mathrm{Mult}(K, \boldsymbol{\pi}_1)$ and $\mathbf{c}_2 \sim \mathrm{Mult}(K, \boldsymbol{\pi}_2)$ with $\lambda \in [0,1]$ satisfy

$$\min_{\tilde{\boldsymbol{\pi}}} \lambda D_{\mathrm{KL}}(\boldsymbol{\pi}_1\|\tilde{\boldsymbol{\pi}}) + (1-\lambda)D_{\mathrm{KL}}(\boldsymbol{\pi}_2\|\tilde{\boldsymbol{\pi}}) \qquad s.t. \sum_{i=1}^{K} \tilde{\pi}_i = 1 \tag{A.4}$$

The Lagrange multiplier form of (A.4) is

$$\mathcal{L}(\tilde{\boldsymbol{\pi}}, t) = \lambda D_{\mathrm{KL}}(\boldsymbol{\pi}_1\|\tilde{\boldsymbol{\pi}}) + (1-\lambda)D_{\mathrm{KL}}(\boldsymbol{\pi}_2\|\tilde{\boldsymbol{\pi}}) + t * (\sum_{i=1}^{K} \tilde{\pi}_i - 1)$$

Table 6: Schedule Parameters

| | MNIST | | SVHN(one-stage) | | SVHN | | Cifar10 | | Cifar100 | |
|---|---|---|---|---|---|---|---|---|---|---|
| | $h_{max}$ | $t_{max}$ | $h_{max}$ | $t_{max}$ | $h_{max}$ | $t_{max}$ | $h_{max}$ | $t_{max}$ | $h_{max}$ | $t_{max}$ |
| $\beta_{\mathbf{z}}$ | 30 | 50 | 1 | 175 | 1e-3 | 150 | 1e-3 | 150 | 1e-1 | 150 |
| $\beta_{\mathbf{c}}$ | 30 | 50 | 1 | 175 | 1 | 150 | 1e-3 | 150 | 1e-3 | 150 |
| $\mathbf{I_z}$ | 17.5 | 50 | 50 | 175 | 1280 | 150 | 200 | 150 | 1280 | 150 |
| $\mathbf{I_c}$ | 17 | 50 | 50 | 175 | 2.3 | 150 | 2.3 | 150 | 4.6 | 150 |
| $w_{M_{\mathbf{z}}}$ | \ | \ | \ | \ | 1e-3 | 150 | 1e-3 | 150 | 1e-1 | 150 |
| $w_{M_{\mathbf{c}}}$ | \ | \ | \ | \ | 1 | 400 | 1 | 280 | 1 | 280 |

and the related KKT conditions are

$$\frac{\partial \mathcal{L}(\tilde{\boldsymbol{\pi}}, t)}{\partial \tilde{\boldsymbol{\pi}}} = t - \frac{\lambda \boldsymbol{\pi}_1 + (1-\lambda)\boldsymbol{\pi}_2}{\tilde{\boldsymbol{\pi}}} = 0$$

$$t * (\sum_{i=1}^{K} \tilde{\pi}_i - 1) = 0$$

(A.5)

Solve the equation (A.5), we can get the closed form of $\tilde{\boldsymbol{\pi}}$ as

$$\tilde{\boldsymbol{\pi}} = \lambda \boldsymbol{\pi}_1 + (1-\lambda)\boldsymbol{\pi}_2 \quad \square$$

## A.5 SCHEDULE ANALYSIS

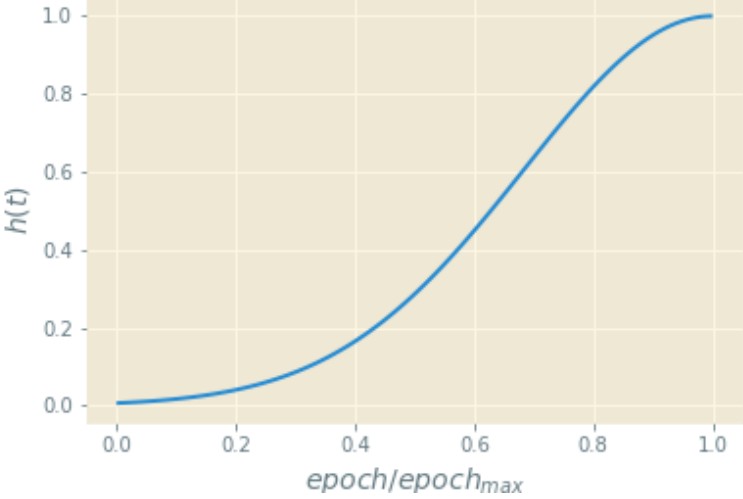

Figure 2: The Exponential Function-Based Scheduler

The warm-up scheduler aims to slowly increase the parameters until they reach their maximum. For a certain hyperparameter $h$, there are 2 parameters control its warm-up process, the target value $h_{max}$ and the total epoch $t_{max}$ to reach the target value. We use the exponential function to get the middle value $h_t$ as

$$h_t = h_{max} \times \exp\left[-5 * (1 - \min(1, t/t_{max}))^2\right]$$

(A.6)

The curve of (A.6) is shown in Figure 2, and we list the scheduler parameters of 4 benchmark datasets, i.e. MNIST, SVHN, Cifar10, Cifar100, in Table 6.

Table 7: Details of Training Process

|  | MNIST | SVHN(one-stage) | SVHN | Cifar10 | Cifar100 |
|---|---|---|---|---|---|
| Latent Dim($\mathbf{z}/\mathbf{c}$) | 10/10 | 32/10 | 128/10 | 128/10 | 128/100 |
| Mutual Info($\mathbf{z}/\mathbf{c}$) | 17.5/17.0 | 50/50 | 1280/2.3 | 200/2.3 | 1280/4.6 |
| loss of $-\log p_{\boldsymbol{\theta}}(\mathbf{X}|\mathbf{c},\mathbf{z})$ | BCE | BCE | MSE | MSE | BCE |
| $\alpha$ of $p_{mixup}(\mathbf{X})$ | \ | \ | 2 | 2 | 2 |
| optimizer |  | Adam |  | SGD | |
| learning rate | 5e-4 | 1e-3 |  | 0.1 | |
| lr scheduler(decay ratio) | \ | \ | every 50 epoch after 200-th(0.5) | [500,600,650](0.2) | |
| weight decay | 0 | 0 |  | 5e-4 | |

## A.6 DETAILS OF TRAINING PROCESS

Here we list some important items need to set in the training process for different process. We classify these items into 2 categories: (1) items related to the loss function and (2) items related to the optimization strategy. The details are as follows and we list the exact value in Table 7:

1. Items related to the loss function

   - Latent dim
     The latent dim of discrete variable $\mathbf{c}$ is the same as the number of classifications, that is, $K$. For continuous variable $\mathbf{z}$, the latent dim is determined by experiments.

   - Mutual information
     We find the value of continuous mutual information will affect the generative performance, but have little impact on semi-supervised learning results, so we choose a suitable value to get the best generative performance. For discrete information, we choose the value in the ideal scene, that is, $\mathbf{I_c} = \log(K)$.

   - Calculation of $\mathbb{E}_{q_\phi(\mathbf{z}|\mathbf{X}),q_\phi(\mathbf{c}|\mathbf{X})} - \log p_{\boldsymbol{\theta}}(\mathbf{X}|\mathbf{c},\mathbf{z})$
     $p_{\boldsymbol{\theta}}(\mathbf{X}|\mathbf{c},\mathbf{z})$ has two forms: (1) normal distribution with $\mathcal{N}(f_\theta(\mathbf{c},\mathbf{z}),\mathbf{I})$ and (2) multinomial distribution with $\text{Mult}(dim(\mathbf{X}),f_\theta(\mathbf{c},\mathbf{z}))$. For the two forms, the loss function of $-\log p_{\boldsymbol{\theta}}(\mathbf{X}|\mathbf{c},\mathbf{z})$ is mean square error(MSE) and binary cross entropy(BCE) respectively. We use reparameterization trick in (20) to approximate expectation, and the sampling frequency is 1.

   - $\alpha$ of $p_{mixup}(\mathbf{X})$
     The $\beta(\alpha,\alpha)$ for mixup vicinal distribution will strongly affect SSL performance. We set it to 2 in all experiments.

2. Items related to the optimization strategy

   - Optimizer
     In one-stage VAE, we use Adam. In OSPOT-VAE, we use SGD with momentum 0.9.

   - Learning rate
     We set the initial learning rate to 0.1 in OSPOT-VAE, which obtains best SSL performance.

   - The scheduler of adjusting learning rate
     We decay the learning ratio in some milestones with the specified decay rate.

   - Weight decay
     Weight decay controls the strength of $L_2$ regularization.

We also list some standard training curves on benchmark datasets in Figure 3 and 4.

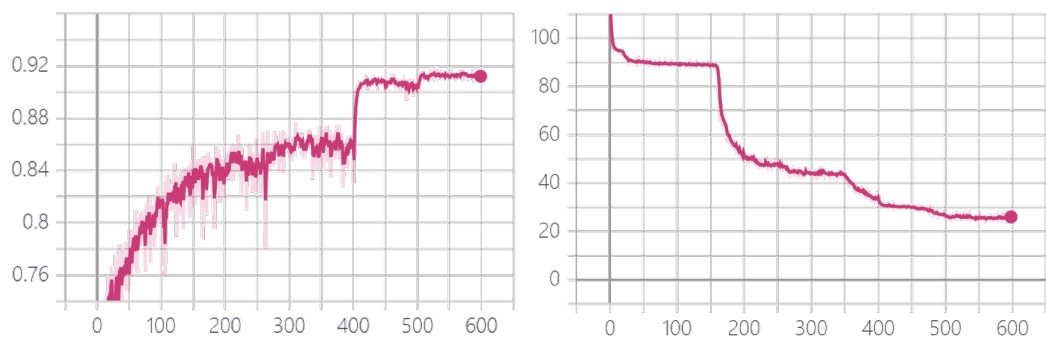

Figure 3: The training curves of Cifar10.
Left: classification performance. Right: $\mathbb{E}_{q_\phi(\mathbf{z}|\mathbf{X}), q_\phi(\mathbf{c}|\mathbf{X})} - \log p_{\boldsymbol{\theta}}(\mathbf{X}|\mathbf{c}, \mathbf{z})$

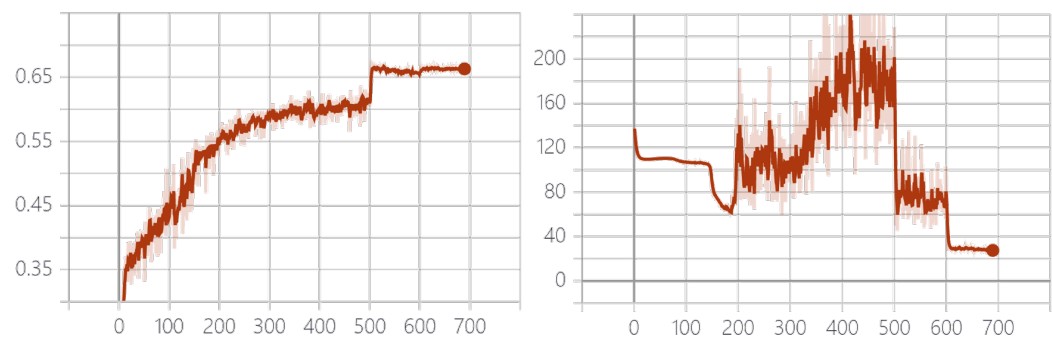

Figure 4: The training curves of Cifar100.
Left: classification performance. Right: $\mathbb{E}_{q_\phi(\mathbf{z}|\mathbf{X}), q_\phi(\mathbf{c}|\mathbf{X})} - \log p_{\boldsymbol{\theta}}(\mathbf{X}|\mathbf{c}, \mathbf{z})$

### A.7 GENERATIVE PERFORMANCE

Following figures5-7 show the generation performance of our OSPOT-VAE.

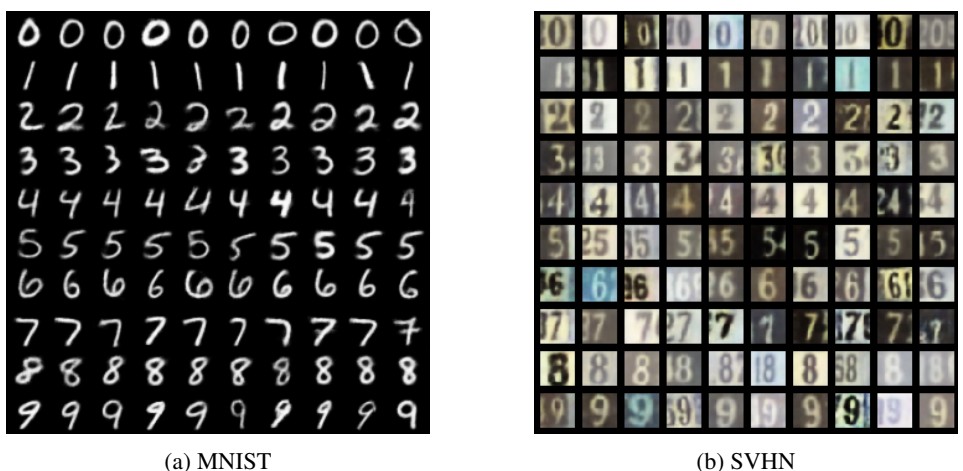

(a) MNIST  (b) SVHN

Figure 5: The generative performance of OSPOT-VAE in MNIST and SVHN

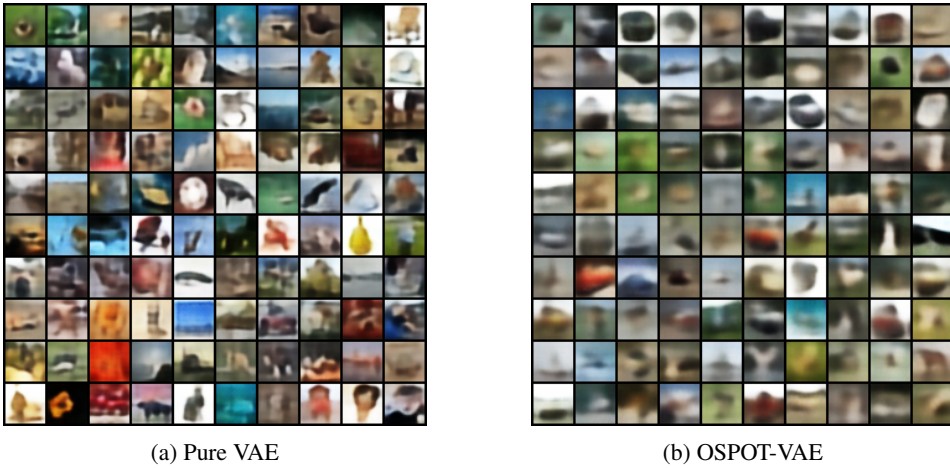

(a) Pure VAE                                        (b) OSPOT-VAE

Figure 6: Compare the generative performance of pure VAE and OSPOT-VAE in Cifar10

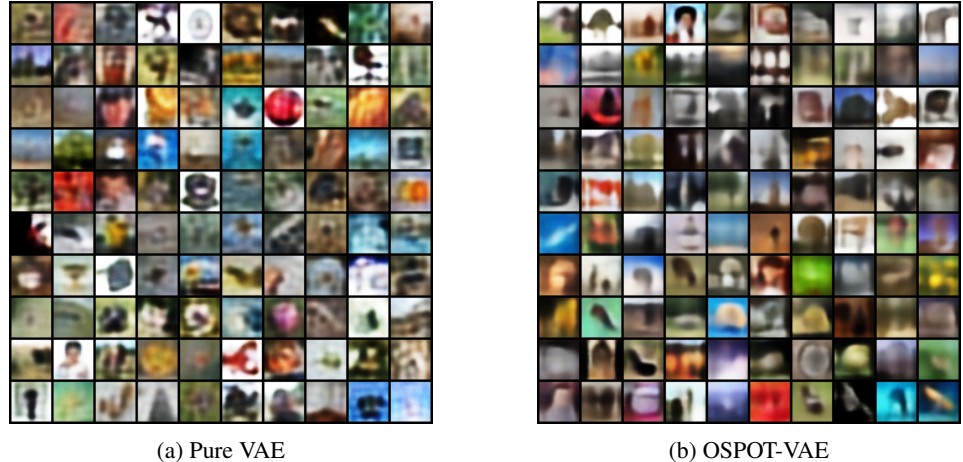

(a) Pure VAE                                        (b) OSPOT-VAE

Figure 7: Compare the generative performance of pure VAE and OSPOT-VAE in Cifar100

