# OpenReview forum: "Good Semi-supervised VAE Requires Tighter Evidence Lower Bound"
_ICLR.cc/2020/Conference — Reject_

### Official Review · AnonReviewer2 · 2019-10-22
**Official Blind Review #2**

**Rating:** 3

**Review:**

This paper aims to solve the existing problems in semi-supervised learning by employing optimal transport theory and proposes a one-stage training VAE, OSPOT-VAE, which has a tighter ELBO and demonstrates better performance on benchmark datasets like CIFAR-10 and CIFAR-100 than benchmark methods. One main contribution in the paper is to use optimal transport to achieve a tighter ELBO in the proposed framework.

My main concern is that it appears in the paper the authors are trying to use a linear transport to approximate the intractable posterior distribution p(z|X) and p(c|X). I am not sure how this can be achieved and there is no discussions in the paper about this or how to reduce the approximation errors.



**Experience Assessment:**

I have published one or two papers in this area.

**Review Assessment: Checking Correctness Of Derivations And Theory:**

I assessed the sensibility of the derivations and theory.

**Review Assessment: Checking Correctness Of Experiments:**

I assessed the sensibility of the experiments.

**Review Assessment: Thoroughness In Paper Reading:**

I read the paper at least twice and used my best judgement in assessing the paper.

---

> ### Author Response · Authors · 2019-11-06
> **Rebuttal Page: some clarities to the pointed out problems**
>
> The authors would like to thank the reviewers for their helpful comments. Please find the attached reviews with our changes and responses. Thank you again for the reviews, they were very helpful for the revision and the time and detail put into these reviews was much appreciated
>
> 1. My main concern is that it appears in the paper the authors are trying to use a linear transport to approximate the intractable posterior distribution $p(z|X)$ and $p(c|X)$.
>
> It seems that the writing of this paper has caused some confusion and we apologize for this. Actually, we aim to derive an estimation of the KL divergence between the intractable posterior distribution $p(z|X)$ and $p(c|X)$ and the inferred distribution $q_{\phi}(z|X),q_{\phi}(c|X)$ as $\mathbb{D}_{KL}(q_{\phi}(z|X)\Vert p(z|X))$ and $\mathbb{D}_{KL}(q_{\phi}(c|X)\Vert p(c|X))$. We use the mixup distribution $p_{mixup}$ instead of the empirical distribution $p_{emp}$ and derive the approximation of these two KL divergences. In experiments, we prove that this approximation achieves a tighter ELBO in MNIST, SVHN, Cifar10 and Cifar100 datasets.
>
> Thanks again for your hard work. We will reorganize the illustration in section 3 and provide a new version soon.

---

### Official Review · AnonReviewer1 · 2019-10-22
**Official Blind Review #1**

**Rating:** 3

**Review:**

The paper proposes to combine a VAE model with the Optimal Transport to approximate some components of the model. The authors evaluate their approach on semi-supervised problems and claim to obtain very competitive results compared to literature. Unfortunately, the paper is very unclear and hard to follow. The authors make some claims that are not true, for instance, learning so called M1+M2 architecture end-to-end is hard, however, there are papers that successfully train such model. Moreover, the authors reported their results as SOTA, however, they missed many other papers with much better scores.

Remarks:
- The authors claim in the introduction that training a two-level VAE with a classifier (so called M1+M2 architecture) is not robust. However, there are papers that were able to train such model without any reported problems, for instance:
* Louizos, C., Swersky, K., Li, Y., Welling, M., & Zemel, R. (2015). The variational fair autoencoder. arXiv preprint arXiv:1511.00830.
* Davidson, T. R., Falorsi, L., De Cao, N., Kipf, T., & Tomczak, J. M. (2018). Hyperspherical variational auto-encoders. arXiv preprint arXiv:1804.00891. (published at UAI 2018)
* Ilse, M., Tomczak, J. M., Louizos, C., & Welling, M. (2019). DIVA: Domain Invariant Variational Autoencoders. arXiv preprint arXiv:1905.10427.

 - The authors report SOTA results on MNIST (100 labels). However, there are papers that report better scores, for instance 2.6 in:
Davidson, T. R., Falorsi, L., De Cao, N., Kipf, T., & Tomczak, J. M. (2018). Hyperspherical variational auto-encoders. arXiv preprint arXiv:1804.00891. (published at UAI 2018)
In this paper they used a VAE model (the M1+M2 architecture) that was trained end-to-end.

- A rather minor remark, but I do not fully see a reason why the authors spent a lot of space on Section 2. I believe most of this text could be included in the Appendix.

- Equation 8: The authors claim that this decomposition was introduced in (Zhao et al., 2017), however, it was done earlier:
Hoffman, M. D., & Johnson, M. J. (2016, December). ELBO surgery: Yet another way to carve up the variational evidence lower bound. In Workshop in Advances in Approximate Bayesian Inference, NIPS.

- Equation (16) is very vague and hard to follow. Moreover, the idea of using Optimal Transport is also hard to follow.  The authors present a very generic description of the Optimal Transport, and then refer to an algorithm (a pseudocode). It is very unreadable.

===== AFTER REBUTTAL =====
I would like to thank the authors for the rebuttal. I read the comments carefully. However, I am not still convinced by claims of the paper. It is still vague to me why the proposed training procedure is necessary. Therefore, I decided to keep my original score.

**Experience Assessment:**

I have published in this field for several years.

**Review Assessment: Checking Correctness Of Derivations And Theory:**

I assessed the sensibility of the derivations and theory.

**Review Assessment: Checking Correctness Of Experiments:**

I assessed the sensibility of the experiments.

**Review Assessment: Thoroughness In Paper Reading:**

I read the paper at least twice and used my best judgement in assessing the paper.

---

> ### Author Response · Authors · 2019-11-06
> **Rebuttal Page: some clarities to the pointed out problems**
>
> The authors would like to thank the reviewers for their helpful comments. Please find the attached reviews with our changes and responses. Thank you again for the reviews, they were very helpful for the revision and the time and detail put into these reviews was much appreciated
>
> 1. The authors claim in the introduction that training a two-level VAE with a classifier (so-called M1+M2 architecture) is not robust. However, there are papers that were able to train such a model without any reported problems
>
> In the simple datasets such as MNIST and Fashion MNIST, the two-stage VAE can be trained well end-to-end. However, when it comes to more sophisticated datasets, the two-stage training strategy is not robust, and this is the main reason why semi-supervised VAE can not obtain success in some real-world datasets such as Cifar10 and Cifar100. Actually, an implicit contribution of our paper is to extend the semi-supervised VAE model to these real-world datasets, which is something no one has done before.
>
> 2. The authors report SOTA results on MNIST (100 labels).
>
> Thank you for pointing this out, this is our negligence. We will add more experiments later. However, we find that in a fair comparison with the same latent variable number ($dim_{z_1}+dim_{z_2}=10$ and $dim_y=10$), the two-stage result in "Hyperspherical variational auto-encoders" is 5.9, which is higher than our result 3.14.
>
> Besides, we mainly declare the SOTA results in Cifar10 and Cifar100.
>
> 3. Equation 8: The authors claim that this decomposition...
>
> Thanks again for pointing this out and we will correct the reference order.
>
> 4. Equation (16) is very vague and hard to follow:
>
> It seems that the writing of this paper has caused some confusion and we apologize for this. Actually, we aim to derive an estimation of the KL divergence between the intractable posterior distribution $p(z|X)$ and $p(c|X)$ and the inferred distribution $q_{\phi}(z|X),q_{\phi}(c|X)$ as $\mathbb{D}_{KL}(q_{\phi}(z|X)\Vert p(z|X))$ and $\mathbb{D}_{KL}(q_{\phi}(c|X)\Vert p(c|X))$. We use the mixup distribution $p_{mixup}$ instead of the empirical distribution $p_{emp}$ and derive the approximation of these two KL divergences. In experiments, we prove that this approximation achieves a tighter ELBO in MNIST, SVHN, Cifar10 and Cifar100 datasets.
>
> Thanks again for your hard work. We will reorganize the illustration in section 3 and provide a new version soon.

---

### Official Review · AnonReviewer3 · 2019-10-22
**Official Blind Review #3**

**Rating:** 3

**Review:**

Contributions:

The main contribution of this paper lies in the proposed OSPOT-VAE for semi-supervised classification. The proposed model is (i) an one-stage framework that unifies the generation and classification loss in one ELBO, (ii) achieves a tighter ELBO by applying optimal transport to the distribution of the latent variables, and (iii) achieves SOTA semi-supervised learning results without sacrificing generative performance on many benchmarks.

Strengths:

(1) Novelty: This paper contains novelty in terms of two aspects. First, the derivation of Eqn. (13) unifies both generation loss and classification loss naturally. The derivation seems different from a standard semi-supervised VAE. Second, the use of optimal transport (OT) estimation to tighten the ELBO results in improved classification performance. The use of OT seems novel in this context.

(2) Writing: The paper is generally well written. However, I also found section 3 is hard to follow, with details in the Weaknesses section.

(3) Experiments: The results look impressive. OSPOT-VAE achieves better performance than other generative approaches.

Weaknesses:

(1) Clarity: My biggest concern lies in the clarity of section 3, which I think is confusing in its current shape. After several rounds of reading, below is my understanding. In section 3.1, the authors first derive the objective in Eqn. (13). This objective is derived based on the use of mutual information (Eqn. (8)). By the end, no special cross-entropy loss term like in Eqn. (7) is needed.  Second, in order to tighten the bound, section 3.2 introduces optimal transport estimation. Combining these two losses together (see in Algorithm 2) results in the final algorithm. However, generally, I feel the presentation of section 3 is confusing. One may need to present the big picture (framework) first, and then dig into details.

Besides that, I have some questions as below.

a) There are many hyper-parameters in the final objective, for example, the mutual information I_z, I_c is also considered as hyper-parameters. How all these hyper-parameters are selected?

b) Is there any intuition on why Eqn. (13) is already enough without introducing the empirical cross-entropy loss like in Eqn. (7)? It seems to me adding this additional cross-entropy loss will not hurt performance.

c) In Eqn. (12), how is the last term D_{KL}(p(c|X)||q_{\phi}(c|X)) calculated, since we cannot get the true p(c|X)?

d) As shown in Table 4, the use of optimal transport seems to be the key. However, section 3.2 is hard to follow. For example, In Algorithm 1, Line 6 & 7 considers the true posterior of z and c is a Gaussian and a multinomial distribution, respectively. Why doing this? Supported by theory? I also feel the calculation of L_{Rz} and L_{Rc} in Line 8 & 9 is hard to follow.

e) How reliable is this optimal transport estimation is? And how it contributes to the final classification performance? Since this loss term is not a classification loss, then is there any intuition on why it serves as a regularization to help the final performance so much?

f) Results in Table 5 on Cifar10 seem not imply OSPOT-VAE tightens the ELBO. Adding results on MNIST will be better, since the ELBO on MNIST is widely benchmarked.

(2) Minor:

a) I am not sure why section 2.1 is needed. Also, the feature matching GAN paragraph in section 2.2 seems also unnecessary. If you really want to discuss the use of GAN for semi-supervised learning, then there is a lot of other work that needs discussion.

b) There is a missing right bracket in the second term of Eqn. (9).

Overall, I think the results look quite impressive. However, the presentation of the method part is unclear to me, and could be much improved.





**Experience Assessment:**

I have published one or two papers in this area.

**Review Assessment: Checking Correctness Of Derivations And Theory:**

I assessed the sensibility of the derivations and theory.

**Review Assessment: Checking Correctness Of Experiments:**

I assessed the sensibility of the experiments.

**Review Assessment: Thoroughness In Paper Reading:**

I read the paper at least twice and used my best judgement in assessing the paper.

---

> ### Author Response · Authors · 2019-11-06
> **Rebuttal Page: some clarities to the pointed out problems**
>
> Thank you for your careful review. These pointed problems are of great value, and we will make further improvements based on them. Here we are trying to clarify some problems above:
>
> Problem 1(Clarity: My biggest concern lies in the clarity of section 3...), (b), (c) :
>
> As you said, we find that our writing order is very confusing, and your understanding is correct. Here we forgot to declare an equivalent form in advance as
> $$
> \text{CrossEntropy }(y\sim p(c|X),q_{\phi}(c|X))=-\log (q_{\phi}(y|X))= \mathbb{D}_{KL}(p(c|X)\Vert q_{\phi}(c|X))  \text{ when }  p(c|X)=\textbf{Mult}(1,C), \text{ and } 1 \text{ is a one-hot vector}.
> $$
> With this equivalent form,  instead of simply adding an extra cross-entropy part in VAE model, we naturally derive the $\mathbb{D}_{KL}(p(c|X)\Vert q_{\phi}(c|X))$ from the ELBO directly. The Equation (12) is based on the labeled pairs $(X,y)\in \mathbb{D}_{L}$, which means the $p(c|X)=\text{Mult}(y,C)$ is known.
>
> I hope the explanations above could answer your problem (1),(b),(c).
>
> Problem 1. (a) :
>
> We should write an appendix to elaborate on this part separately, which is our negligence. The selection of the mutual information $I_{c}$ is very simple, for the classification prior $p(c)$ is known or can be simply estimated, and the best inference of $ q_{\phi}(c|X)$ should be a one-hot vector $\text{Mult}(1, C)$. In this optimal situation, $I_{c}$ can be easily calculated, e.g. $I_{c}=\log 10$ in Cifar10 and $I_{c}=\log 100$ in Cifar100. The selection of $I_{z}$ is a little tricky, and we choose $I_{z}$ in a totally empirical unsupervised process, making $I_{z}$ as
> $$
> \min_{I_z}\max_{\theta,\phi} -\log(p(X|z))
> $$
>
> Problem 1.(d),(e) :
>
> In section 3.2, we explain that when ELBO is good enough, continuing optimization ELBO may not result in a better semi-supervised representation, which means the margin between the true likelihood $\log p(x)$ and the ELBO
>
> $$
> \mathbb{D}_{KL}(q_{\phi}(z|X)\Vert p(z|X))+\mathbb{D}_{KL}(q_{\phi}(c|X)\Vert p(c|X)) =\log p(X)-ELBO
> $$
>
> can be very large. In this situation, we use the $p_{mixup}$ distribution as well as the optimal transport scheme in latent space to create the estimation of $\mathbb{D}_{KL}(q_{\phi}(z|X)\Vert p(z|X))$ and $\mathbb{D}_{KL}(q_{\phi}(c|X)\Vert p(c|X))$. To minimize this estimation, we get a better model with tighter ELBO, which is proved with several experiments in benchmark datasets (Cifar10, Cifar100, and SVHN) in ablation study. The better semi-supervised accuracy means the lower result of $\mathbb{D}_{KL}(q_{\phi}(c|X)\Vert p(c|X))$, which proves the tighter ELBO assumption.
>
> Problem 1.(f) :
>
> We will add experiments and report our results in MNIST soon.
>
> Problem 2.(a) :
>
> Our method achieves the best performance among all generative models, which can be separated into two groups: VAE and GAN. For this purpose, we selectively introduce some SOTA semi-supervised GAN models.
>
> Thanks for your hard work again! We will reorganize the illustration of Section 3 and provide a new version soon.

---

### Author Response · Authors · 2019-10-21
**Modify some symbol errors and display errors in Figure 1: The schematic of OSPOT-VAE**

We have modified a symbol error and a display error in Figure 1: The schematic of OSPOT-VAE.

1. Change the original bitmap figure to the vector graph (from .png to .svg).

2. Change the KL-divergency symbol 'KL()' to 'D_{KL}()', matching the symbols in the paper.

The new figure is now available on the code page [ https://github.com/PaperCodeSubmission/OSPOT-VAE ], and we will fix it in the article when "paper_diff" is allowed to be submitted.

---

### Author Response · Authors · 2019-11-05
**Submit a new paper version**

We have submitted a new paper version named "paper_diff", which can be obtained at https://github.com/PaperCodeSubmission/OSPOT-VAE/blob/master/paper_diff.pdf . Following are the main changes in this new version:

1. We have provided a more clear and explainable schematic of OSPOT-VAE in figure 1.

2. We have modified the illustration of the optimal transport scheme on page 6 and changed some ambiguous concepts.

3. We have modified some symbols and display errors, e.g. using $L_{M_{z}}$ instead of $L_{R_{z}}$ to represent the estimation of the margin.

We sincerely appreciate the hard work of reviewers for taking the time to review the papers during your own deadlines.  We hope to get your valuable reviews and suggestions soon.

---

### Decision · Program_Chairs · 2019-12-19

**Decision:**

Reject

**Comment:**

The paper proposes to combine a VAE model with the Optimal Transport to approximate some components of the model. The authors evaluate their approach on semi-supervised problems and claim to obtain very competitive results compared to literature. Unfortunately, the paper would benefit substantially from revisions to make it easier to follow. For this reason, the paper is not ready for publication in this venue at this time.